# Preliminary Study of the Antimicrobial, Anticoagulant, Antioxidant, Cytotoxic, and Anti-Inflammatory Activity of Five Selected Plants with Therapeutic Application in Dentistry

**DOI:** 10.3390/ijerph19137927

**Published:** 2022-06-28

**Authors:** Sonia M. López Villarreal, Joel H. Elizondo Luévano, Raymundo A. Pérez Hernández, Eduardo Sánchez García, María J. Verde Star, Roció Castro Ríos, Marsela Garza Tapia, Osvelia E. Rodríguez Luis, Abelardo Chávez Montes

**Affiliations:** 1Facultad de Odontologia, Universidad Autonoma de Nuevo Leon, Monterrey 64460, Mexico; sonia.lopezvl@uanl.edu.mx; 2Facultad de Ciencias Biologicas, Universidad Autonoma de Nuevo Leon, San Nicolás de los Garza 66455, Mexico; joel.elizondolv@uanl.edu.mx (J.H.E.L.); raymundo.perezhrz@uanl.edu.mx (R.A.P.H.); eduardo.sanchezgrc@uanl.edu.mx (E.S.G.); maria.verdest@uanl.edu.mx (M.J.V.S.); 3Facultad de Medicina, Universidad Autonoma de Nuevo Leon, Monterrey 64460, Mexico; rocio.castrors@uanl.edu.mx (R.C.R.); marsela.garzatp@uanl.edu.mx (M.G.T.)

**Keywords:** antibacterial activity, anticoagulant activity, anti-inflammatory properties, interleukins, medicinal plants, natural extracts, traditional medicine

## Abstract

The usefulness of traditional plants in Mexico to treat human ailments has been known since ancient times. This work evaluated the antimicrobial, anticoagulant, antioxidant, cytotoxic, and anti-inflammatory potential of ethanolic extracts of *Aloe vera*, *Equisetum arvense*, *Mimosa tenuiflora*, *Lippia graveolens*, and *Syzygium aromaticum*. The antimicrobial activity of the extracts was evaluated against *Streptococcus mutans* and *Streptococcus sorbinus*; a significant inhibitory effect of the *L. graveolens* extract on both bacteria was observed at concentration levels of 250 µg/mL and greater. The anticoagulant activity was evaluated in terms of prothrombin time (PT) and activated partial thromboplastin time (APTT), *A. vera* and *M. tenuiflora* extracts showed no significant difference (*p* ˂ 0.05) in PT compared with the control, and for APTT the extracts of *A. vera*, *L. graveolens*, and *S. aromaticum* decreased the APTT significantly (*p* ˂ 0.05) compared with the control. The antioxidant potential by DPPH assay indicated that the *E. arvense* extract behaved statistically the same as the control. The cytotoxic activity was evaluated in HGF-1 cells using the fluorometric microculture cytotoxicity assay technique, and none of the extracts was toxic at 125 and 250 µg/mL concentrations. Finally, the anti-inflammatory activity was evaluated using ELISA, where the *A. vera* extract showed the best anti-inflammatory capacity. Further research on the search for bioactive metabolites and elucidation of action mechanisms of the most promising extracts will be carried out.

## 1. Introduction

The damage or loss of primary teeth causes severe problems for children such as the tilting of teeth, reduction in the vertical dimension in the case of molars, closed bite in the case of incisors, loss of esthetics, psychological disorders, language problems, speech defects, as well as social effects, among others; therefore, preserving a primary tooth is significant [1]. One of the leading causes of the loss of primary teeth at an early age is oral caries [2]. From the large number of bacteria found in the oral cavity, microorganisms belonging to the genus *Streptococcus*, specifically the *mutans* and *sobrinus* species, have been associated with caries [3]. Different studies indicate that the incidence of caries is higher when *S. mutans* and *S. sobrinus* coexist in the oral microflora. Therefore, they should be considered equally virulent concerning dental caries [4].

Caries can cause pulp exposure accompanied by acute or chronic pulp infection and therefore can cause pain to the patient; for these reasons, immediate attention should be given to a primary tooth with pulp exposure, pulpotomy being the treatment of choice to preserve them [5]. Pulpotomy treatment in primary teeth consists of the removal of the pulp located in the crown or coronal pulp, the placement of a medication such as formocresol at the entrance of the root canals to preserve the root pulp, and the final permanent restoration, which is traditionally a steel crown [5].

The material or substance placed in contact with the root pulp must meet the ideal conditions of being bactericidal, harmless to the pulp and surrounding tissues, and promoting the preservation and repair of the root pulp [6]. The common treatment is formocresol which has bacteriostatic and coagulant properties; although it has been the treatment of choice for decades, it has been associated with pulp devitalization [7]. On the other hand, it has been proven to have toxic effects, has been considered a potential carcinogen, and has presented cytotoxicity, mutagenicity, and embryotoxic and teratogenic effects [7]. Therefore, it is necessary to find new alternatives such as natural products like plants [8]. Plants are a source of bioactive molecules because they have properties reported as antimicrobial, antiparasitic, anticancer, healing, analgesic, anti-inflammatory, and tissue repairing, among others, and represent a potential alternative in pulp treatment [9,10,11]. The discovery and use of natural sources such as plants are not new because approximately 80% of the world’s population in developing countries uses plants to treat various ailments [12]. This is mainly due to the presence of adverse effects or drug resistance by microorganisms, so the increase in preference for plant-based products is increasing daily. However, many studies are still needed to explore the potential use of traditional or indigenous medicinal plants to treat diseases [13].

Among the plants commonly used to treat oral or dental problems, *Aloe vera* (*Xanthorrhoeaceae*) can reduce plaque, gingivitis, and *S. mutans* in the oral cavity with effects comparable to those of chlorhexidine [14]. One of the most widely used plants worldwide to treat various conditions is *Equisetum arvense* (*Equisetaceae*), which is a plant that has analgesic, anti-inflammatory action comparable to that of conventional drugs against osteoarthritis or rheumatoid arthritis and has efficacy against pain; however, its mechanisms of action are not yet entirely known [15]. *Mimosa tenuiflora* (*Leguminosae*) is a plant that possesses flavonoids and tannins with a high potential for controlling microbial growth on bacteria of medical importance such as *Escherichia coli*, *Enterobacter aerogenes*, *Pseudomonas aeruginosa*, *Klebsiella pneumoniae*, *Staphylococcus aureus*, and *Providencia* spp [16]. Teas or infusions of *Lippia graveolens* (*Verbenaceae*) have shown activity against microbes related to oral health and hygiene, such as *Staphylococcus aureus*, *E. coli*, *S. mutans*, *Lactobacillus acidophilus,* and *Candida albicans* [17]. Another of the most commonly used plants for oral problems is *Syzygium aromaticum* (*Myrtaceae*). The essential oil eugenol is widely used as an anesthetic treatment for toothache and has anti-inflammatory and antimicrobial properties [18].

Therefore, this study focused on the antimicrobial, anti-inflammatory, coagulating, and antioxidant capacity of ethanolic extracts from the five plants mentioned above, commonly known as aloe or sabila, horsetail tepezcohuite, Mexican oregano, and clove, respectively, as an alternative treatment for the pulpotomy. These plants were selected for their high demand for use as traditional medicinal agents in Mexico and Latin America, as well as for their previously mentioned scientific properties. For this reason, the relevance of this study is focused on demonstrating the beneficial potential of these plants as well as providing a scientifically focused overview of their safe use.

## 2. Materials and Methods

### 2.1. Chemicals and Reagents

All chemicals and solvents were of analytical grade. 1,1′-Hexamethylenebis[5-(4-chlorophenyl)biguanide] (Chlorhexidine), 2,2-diphenyl-1-picrylhydrazyl (DPPH), absolute ethanol (EtOH), amphotericin B, antibiotic-antimycotic solution (penicillin/streptomycin) stabilized, dimethyl sulfoxide (DMSO), Dulbecco’s Modified Eagle Medium (DMEM), fetal bovine serum (FBS), fluorescein diacetate (FDA), l-glutamine, thiazolyl blue tetrazolium bromide (MTT), and vitamin E were purchased from Sigma-Aldrich (Merck KGaA, Darmstadt, Germany). The Interleukin 1 Beta (IL-1β) human, Interleukin 10 (IL-10) human, and Tumor necrosis factor-alpha (TNF-α) human ELISA kits were purchased from Invitrogen™ (Thermo Fisher Scientific Inc., Madrid, Spain). Muller-Hinton (MH) agar medium was purchased from BD Bioxon (Becton Dickinson & Company, Franklin Lakes, NJ, USA).

### 2.2. Cell Culture

Human gingival fibroblasts (HGF-1, ATCC CRL-7222) were maintained in DMEM, supplemented with 10% FBS, 2 mM L-glutamine, 100 units/mL penicillin/streptomycin, and 0.25 μg/mL amphotericin B at 37 °C, 5% CO_2_, and 95% relative humidity [19].

### 2.3. Plant Material, Extraction, and Phytochemical Tests

The plants (Table 1) were purchased in Pacalli^®^ (www.pacalli.com.mx, accessed on 12 February 2022), a local store in the city of Monterrey, Mexico, in dried form and were identified by Professor Dr. Marco A. Guzman Lucio, Chief of the Herbarium of Facultad de Ciencias Biologicas at Universidad Autonoma de Nuevo Leon, Mexico. The extraction was performed with EtOH. For this, 60 g of milled plant material was subjected to extraction with 600 mL of EtOH, each using Soxhlet equipment for 48 h [20]. Before extraction, each specimen was ground using a manual mill, being careful not to pulverize them. Finally, the extracts were filtered, and EtOH was removed in a rotary evaporator under reduced pressure and stored (4 °C) in amber bottles until use. Equation (1) was used to calculate the extraction yield percentage:(1)$$\%\;\mathrm{Yield}=\left(\;\frac{\mathrm{Final}\;\mathrm{weight}}{\mathrm{Initial}\;\mathrm{weight}}\right)\times 100$$

The following tests were performed to identify the phytochemical groups of each extract: KMnO_4_ (unsaturations), Antrone (carbohydrates), Lieberman–Buchard (sterols, triterpenes), Shinoda (flavonoids), NaOH (coumarins), Baljet (sesquiterpene lactones), sulfuric acid (quinones), NaHCO_3_ (carboxyl group), ferric chloride (tannins) and Dragendorff (alkaloids) [21,22,23].

### 2.4. Antimicrobial Assays

The antimicrobial activity of the extracts was evaluated against *S. mutans* (ATCC 700611) and *S. sobrinus* (ATCC 700611) by 0.05 on the McFarland scale. The strains *S. mutans* and *S. sobrinus* were chosen because they are the main microorganisms causing caries and pulp pathologies, including pulpitis. The agar-well diffusion method was used to investigate antimicrobial activity [24]. Autoclaved Mueller–Hinton agar medium (20 g/L; pH 7) was poured into Petri plates, and overnight, refreshed bacterial culture was inoculated. Wells (10 mm) were labeled, and plant extracts (250 to 3000 µg/mL), 0.12% chlorhexidine (C+: positive control), and EtOH (negative control) were poured into their respective wells. The inhibition zone (mm) was measured after 24 h incubation at 37 °C.

### 2.5. In Vitro Anticoagulant Assays

The coagulant activity of the extracts was analyzed to verify that the extracts did not interfere with the coagulation process. Prothrombin time (PT) and activated partial thromboplastin time (APTT) were measured by ascertaining the behavior of the extract in human venous blood samples from healthy donors. Peripheral venous blood samples were drawn from the donor with clotting times within the normal range using 4.5 mL BD Vacutainer^®^ (Becton, Dickinson & Company, NJ, USA) tubes with 3.2% sodium citrate anticoagulant. The venous blood sample was centrifuged at 5000 rpm for 15 min, then 2.9 mL of plasma plus 100 µL of the corresponding extract were placed in microcentrifuge tubes, and the final concentrations for each extract evaluated were 500 and 1000 µg/mL; for the blank, no extract was added, only 3 mL of plasma. The tubes were incubated for 4 min at 37 °C [25]. The test was performed using an automated coagulometer (Sysmex CA-560; Sysmex, Kobe, Japan); the instrument marks precisely the moment the clot is produced, and the results are expressed in seconds.

### 2.6. Antioxidant Activity Tests

The plant extracts’ antioxidant capacity at different concentrations (31.25 to 500 µg/mL) was tested by using a DPPH radical assay [26]. The DPPH was prepared as 125 μM in methanol, 100 μL of each sample was taken, and 100 μL of DPPH was added. The samples were incubated at 37 °C room temperature for 30 min, protected from light. The absorbance (Abs) at 517 nm was measured using a spectrophotometer Genesys 20 (Thermo Fisher Scientific, Waltham, MA, USA). The inhibitory percentage of the DPPH radical scavenging ability was calculated against the control of the DPPH solution without a sample. As a positive control (C+), a solution of vitamin C was used and EtOH as a blank control; the reduction percentage was calculated using Equation (2):(2)$$\%\;\mathrm{DPPH}\;\mathrm{Reduction}=\left(\;\frac{\mathrm{Abs}\;\mathrm{Blank}-\mathrm{Abs}\;\mathrm{Treatment}}{\mathrm{Abs}\;\mathrm{Blank}}\right)\times 100$$
where Abs Blank: Absorbance of the DPPH (DPPH without sample), Abs Treatment: absorbance of the reaction mixture (DPPH with sample).

### 2.7. Cytotoxic Activity Assays

Cell viability on HGF-1 cells was measured using a fluorometric microculture cytotoxicity assay (FMCA) [27]. First, 2 × 10^4^ cells per well were placed in a 96-well microplate with DMEM medium plus 10% FBS at 37 °C (5% CO_2_) and an atmosphere of 95% humidity [19]. These were left to incubate for 24 h for adaptation. Extracts were added at different concentrations (250–1000 μg/mL) and incubated for 24 h, after which the medium was removed, and the cells were washed with phosphate-buffered saline (PBS) [28]. Next, 100 μL of PBS containing 10 μg/mL FDA was added, followed by 30 min of incubation, after which fluorescence was measured at 495 nm in a scanning fluorometer, GloMax^®^ Multi + Microplate Multimode (Promega, Madison, WI, USA). The negative control (C−) consisted only of culture medium. Viability was quantified as a function of the relative fluorescence intensity proportional to the number of cells that survived. The values were standardized with the relative fluorescence units of the C− conditions. The percentage of live cells under experimental conditions was obtained from this value. Cell viability was determined using Equation (3):(3)$$\%\;\mathrm{Cell}\;\mathrm{viability}=\frac{\mathrm{Abs}\;\mathrm{Treatment}}{\mathrm{Abs}\;C-}\times 100$$

The cytotoxicity parameters were taken based on the international standard ISO 10993-18:2021 for the biological evaluation of medical devices (Part 18: Chemical characterization of medical device materials within a risk management process). The cytotoxicity classification based on the % of cell viability was as follows. Noncytotoxic: 100–75%, Slightly cytotoxic: 74–50%, moderately cytotoxic: 49–25% and extremely cytotoxic 24–0%.

### 2.8. Anti-Inflammatory Assays

According to the manufacturer ’s recommendations, the levels of interleukin IL-1β, IL-10, and TNF-α on HGF-1 cells were estimated using the Novex Protein life commercial ELISA kits. The absorbance of all samples was read at 450 nm. The levels of cytokines were expressed as pg/mL at 0, 24, 48, and 72 h. The protocol indicated in the kit insert was followed with a sensitivity better than 60 pg/mL, and then the readings were taken at 450 nm.

### 2.9. Statistical Analysis

All results were expressed as mean value ± standard deviation of triplicate determinations from three independent experiments. One-way ANOVA performed the statistical analysis at a 95% confidence level (95% CL), and a Tukey post hoc test was applied. The half-maximal inhibitory concentration (IC_50_) and the upper (UL) and lower (LL) limit values were calculated via the Probit test. Statistical analyses were performed using the IBM–SPSS software Ver. 21 (IBM Corp., New York, NY, USA).

## 3. Results and Discussion

### 3.1. Phytochemical Tests

The results for the phytochemical tests and extraction yields are shown in Table 2. All extracts were positive for unsaturations, and *S. aromaticum* was negative for carbohydrates. The *A. vera* extract was negative in the Lieberman–Burchard test and positive in the NaHCO_3_ test. All extracts except *M. tenuiflora* were negative in the Dragendorff test. Interest in the use of medicinal plants such as those studied in this research has been increasing in Latin America [29,30] because they grow mainly in tropical and subtropical climates, and they contain high contents of phytochemical constituents such as polysaccharides, different essential minerals, amino acids, vitamins, flavonoids, alkaloids, and other active compounds [31,32]. For this reason, they have been applied since pre-Hispanic times as classical medicinal herbs and are used in the cosmetic, food, and health industries [33,34,35].

It is important to note that the presence of some of the metaboitins may vary in the extracts when compared to the literature; this may be due to the concentration of secondary metabolites in the extracts, the type of extraction, the solvent used, and the methodology used to identify them, other important factors are the date and area of collection of the plant, as this may cause diversity in the concentration of these components [36]. In this research, we only qualitatively determined the families of phytochemical compounds in the extracts evaluated; however, analytical studies are needed to qualitatively determine the compounds in each of the extracts as well. It is important to study the mechanisms of action of the active compounds in the plants evaluated in this research, as well as to determine how they exert their biological effects in in vitro and later in vivo models.

### 3.2. Antimicrobial Activity of Extracts

The agar well diffusion method was used to investigate the antimicrobial activity. It was observed that all five extracts showed activity against *S. mutans* at 3000 µg/mL (*p* < 0.05). The extract with the best activity was *L. graveolens* because it showed activity at all the concentrations evaluated in comparison with the positive control, and the one with the lowest activity was the *A. vera* extract because it only showed activity at the highest concentration; the extracts *E. arvense*, *M. tenuiflora*, and *S. aromaticum* showed significant activity (*p* < 0.05) at 1000 µg/mL and above (Table 3). Table 4 shows the results against *S. sorbinus*. The *E. arvense* extract showed no significant activity compared to the positive control. The extract with the best antimicrobial activity was *L. graveolens* with significant activity from 500 µg/mL, followed by *A. vera* with significant activity from 1000 µg/mL. *S. aromaticum* and *E. arvense* extracts showed significant activity at 2000 and 3000 µg/mL, respectively.

In both cases, the ethanol extract of *L. graveolens* was the most effective and showed significant activity compared to the positive control. As shown in Table 3, *L. graveolens* extract showed highly significant activity (*p* < 0.01) compared to C+ at all concentrations tested against *S. mutans*, and Table 4 shows that *L. graveolens* extract showed highly significant activity (*p* < 0.01) at 2000 and 3000 µg/mL compared to C+, when tested against *S. sorbinus*.

Among the measures for chemical control of dental plaque, mouthwashes are used as adjuvants because they interfere with the composition and metabolism of the biofilm [37]. Among the most widely used is chlorhexidine. It is considered the “gold standard” due to its broad-spectrum antimicrobial activity against different oral pathogens such as *S. mutans*, *S. oralis*, *Lactobacillus acidophilus*, *L. fermentum,* or *Candida albicans* [38,39]. However, side effects have limited its applications. Research using herbal mouthwashes such as aloe vera and tea tree oil indicated that they could decrease plaque, gingivitis, and the concentration of *S. mutans* in the oral cavity of children. This activity was comparable to that of chlorhexidine [14]. Therefore, the current trend has focused on searching for natural solutions [39,40].

### 3.3. Anticoagulant Activity of Extracts

With the continuous increase in the search for new treatments from medicinal plants [25], we set out to examine the anticoagulant potential of *A. vera*, *E. arvense*, *M. tenuiflora*, *L. graveolens*, and *S. aromaticum*. Coagulation tests analyzed the coagulant activity of the extracts to verify that the extracts did not interfere with the coagulation process. Therefore, in vitro coagulation tests of PT and APTT were performed and compared with the times of a healthy patient (control patient). Data represent means ± SD in seconds, taking the times of a healthy patient as a positive control.

Figure 1 shows the activity of the extracts versus PT; we observed that the extracts, *E. arvense* at 500 and 1000 µg/mL, *L. graveolens* at 500 and 1000 µg/mL, and *S. aromaticum* at 500 and 1000 µg/mL presented a statistical difference (*p* < 0.05) in comparison with the evaluated control, the other treatments did not present a statistically significant difference (*p* > 0.05) compared with the control. Figure 2 shows the activity of the extracts against APPT. It is observed that the extracts corresponding to *A. vera* at 500 and 1000 µg/mL, *E. arvense* at 1000 µg/mL, *L. graveolens* at 500 and 1000 µg/mL, and *S. aromaticum* at 1000 µg/mL showed significant differences (*p* < 0.05) compared with the control (32.63 s), and the *M. tenuiflora* extract at 500 µg/mL does not show statistically difference (*p* > 0.05) compared with the control. These results indicate that some of the extracts evaluated can improve PT and APTT.

At the time of this research, no studies of the anticoagulant activity of extracts of *A. vera*, *M. tenuiflora*, and *L. graveolens* were found, so we can infer that this is the first research where these plants are evaluated on PT and APPT. In the literature, there is only one record of the anticoagulant activity of *E. arvense*, and *S. aromaticum*. In a previous study, the anticoagulant capacity of *E. arvense* was demonstrated [41]. Our study observed that this extract increases PT and APPT significantly, as the control evaluated. In a 2020 study, the synthesis of silver nanoparticles with chitosan was performed using an ethanolic extract of *S. aromaticum*, where the antimicrobial activity against *Staphylococcus aureus* (MRSA) and vancomycin-resistant *S. aureus* (VRSA), and their anticoagulant and cytotoxic activity were evaluated. The results showed highly significant activity against these bacteria and increased bleeding and thromboplastin. In contrast, the increase in prothrombin and activated partial thromboplastin time was significant at 0.025 and 0.05 mg/kg doses. A considerable reduction in fibrinogen levels was also observed, in addition to which the cytotoxicity of the nanoparticles showed low toxicity [42]. Therefore, our research findings will open a new perspective for developing phytomedicine and its future clinical application.

### 3.4. Antioxidant Activity of Extracts

Antioxidant activity was evaluated using the DPPH radical scavenging method (Table 5). The Probit test was used to determine the concentration to inhibit 50% (IC_50_) of the DPPH radical. Tukey’s test showed that the IC_50_ of C+ (2.23 μg/mL) was not significantly different (*p* > 0.05) from that of the *E. arvense* extract (2.73 μg/mL), the other extracts behaved as follows < *A. vera* (8.66 μg/mL) < *L. graveolens* (19.80 μg/mL) < *S. aromaticum* (37.20 μg/mL) < *M. tenuiflora* (464.06 μg/mL). Antioxidants are essential substances because they can protect the body from damage caused by oxidative stress induced by free radicals [43]. Plants possess bioactive components such as phenolic compounds, flavonoids and tannins, as well as vitamins, minerals, and sugars, which provide antioxidant protection by eliminating free radicals, protecting against DNA damage and even possess anticarcinogenic capacity by acting as inducers of apoptosis [44]. The antioxidant capacity of the evaluated extracts may be because organic or hydroalcoholic extracts have shown a higher antioxidant activity associated with the solvent because these extracts have shown high concentrations of total phenolic compounds [45]. Plant polyphenols act as reducing agents and antioxidants by the hydrogen donating property of their hydroxyl groups; these were found in different plants in the following order leaves > flower > root > stem > fruit [46].

### 3.5. Cytotoxic Activity of Extracts

Cell viability was measured in human gingival fibroblasts using the FMCA assay. Table 6 shows the behavior of the extracts concerning the control evaluated in percent cell viability and the IC_50_ corresponding to each extract evaluated. It is observed that the five extracts at 150 and 250 μg/mL do not present toxicity concerning the criteria mentioned above. At 500 μg/mL, all are above 50% cell viability. At 1000 μg/mL, *E. arvense* and *S. aromaticum* present cell viability lower than 24% at 1000 μg/mL, which is considered extremely cytotoxic. Statistically, the *A. vera* extract did not present a significant difference (*p* ˂ 0.05) in comparison with the control, which indicates that it is not toxic. As for the *M. tenuiflora* and *L. graveolens* extracts, they behaved statistically similar to each other, and when the Tukey test was performed, it was observed that, in comparison with the control group, they behaved in the same way in both cases. Several natural products are widely used for various therapeutic applications due to their safe medicinal properties. Recent studies provide data on outcomes from natural sources with chemotherapeutic potential and without cytotoxic activity [28,47], as observed in the results.

There is evidence of cytotoxic effects on certain cell types with these plants [48]. Hence, there is a great demand for comprehensive toxicity testing before application, especially to determine effective and critical toxicological values, because this represents a key point for future therapeutic applications [31]. So far, in Mexico, out of 300 species belonging to 90 botanical families reported, only 181 species have been experimentally analyzed for their cytotoxic activity, and the other 119 species have been used for the empirical treatment of diseases mainly related to cancer, so it is of utmost importance to carry out scientific studies against different cellular and non-tumoral lines [48]. In addition, more clinical trials focusing on in vivo experiments are needed in the context of the immunomodulatory properties of these natural extracts.

### 3.6. Anti-Inflammatory Activity of Extracts

The results in Table 7 show the cytokine expression levels in human gingival fibroblasts analyzed using ELISA assays upon applying the different extracts at different times at a 500 μg/mL concentration. The results show that TNF-*α* concentrations decrease gradually with *A. vera*, *M. tenuiflora*, and *S. aromaticum* extracts, suggesting a favorable response during acute inflammation [49]; the other extracts cause an increase in this cytokine, indicating that they are pro-inflammatory [50]. TNF-*α* is very important because it has pathophysiological effects, as it is secreted in large quantities in acute and chronic diseases, sepsis, chronic infections, chronic inflammation, and cancer [51]. Different plant extracts from the *Xanthorrhoeaceae* and *Fabaceae* families, such as *A. vera* and *M. tenuiflora,* respectively, have anti-inflammatory and analgesic potential attributed to the various bioactive compounds they contain, such as flavonoids, tannins, and phenols. These phytochemical components could inhibit several biological processes, such as the expression of cyclooxygenase-2, inducible nitric oxide synthase (iNOS), 5-lipoxygenase biosynthesis, and TNF-α [52,53,54]. Regarding IL-1β, which is produced by leukocytes and is essential during inflammatory and infectious processes, it was observed that *A. vera*, *E. arvense*, and *M. tenuiflora* extracts increase the concentration of this cytokine, which indicates its pro-inflammatory activity [55], this process is vital during the process of acute inflammation in pulp treatment [56]. This cytokine helps macrophages, another type of white blood cell, fight infection and allows leukocytes to pass through the blood vessels’ walls and reach the infection sites [55]. Finally, the concentration of IL-10, which is an anti-inflammatory cytokine, was increased by *E. arvense*, *M. tenuiflora,* and *L. graveolens* extracts, which is favorable for reducing the inflammatory process [11] and in tissue repair in the treatment of pulp [57]. IL-10 reduces inflammation by preventing immune cells from producing cytokines [58]. Different studies indicate immunomodulatory action on B and T lymphocytes and the anti-inflammatory action of ***E. arvense***, which is attributed to kynurenic acid because it was proposed as a possible mediator of the anti-inflammatory effect. Kynurenic acid is also an endogenous oxidative metabolite of tryptophan, with glutamate receptor antagonist activity, which may partly explain its analgesic properties [15]. IL-10 in the presence of *S. aromaticum* extract increased at 48 h, but at 72 h, it decreased (Table 7). The ability of *S. aromaticum* to regulate the interleukins evaluated could be caused by eugenol because our results agree with previous studies in which eugenol was evaluated as the main metabolite of *S. aromaticum*, which presented many pharmacological properties such antioxidant, anti-inflammatory, and pro-inflammatory ones. It was demonstrated that eugenol inhibits the expression of iNOS and the release of TNF-α in mouse macrophages and stimulates the production of IL-6 [59]. Furthermore, the analgesic effect of eugenol against toothache and joint pain through the activation of chloride and calcium channels in ganglion cells has been documented [60].

Although further preclinical studies still need to be conducted, these results demonstrate that medicinal plant-based extracts have potent anti-inflammatory activity and thus justify their use in traditional medicine to treat body pain and other inflammation-related diseases and provide a basis for future clinical studies and possible drug development.

## 4. Conclusions

The search for new alternatives with biological activity in medicinal plants represents an alternative for discovering new options for controlling dental diseases, because medicinal plants represent a biotechnological source of bioactive metabolites; however, the limitation could be the concentration of these metabolites present in these plants as well as their isolation and purification. Concerning the results found in the present investigation, it can be concluded that the ethanol extract of *L. graveolens* was the most effective against *S. mutans* and *S. sorbinus*; however, the *A. vera* extract proved to be the extract with the highest antioxidant capacity and better cell viability and anti-inflammatory activity, indicating its possible use for less toxic oral treatments. However, further studies are needed to identify metabolites with biological activity and their molecular mechanisms of action in order to develop a formulation based on natural products as a therapeutic alternative in pulpotomy.

## 5. Future Perspectives

The biological evaluation of plant extracts of traditional use reveals their potential to identify new natural compounds with immunomodulatory, antimicrobial, anticoagulant, and antioxidant effects; however, this is a preliminary study, and more research is needed. Therefore, in the second stage of the present project, we intend to continue with the biodirected fractionation of the most active extracts and the purification of the components with such biological activity and continue determining their molecular mechanisms of action.

## Figures and Tables

**Figure 1 ijerph-19-07927-f001:**
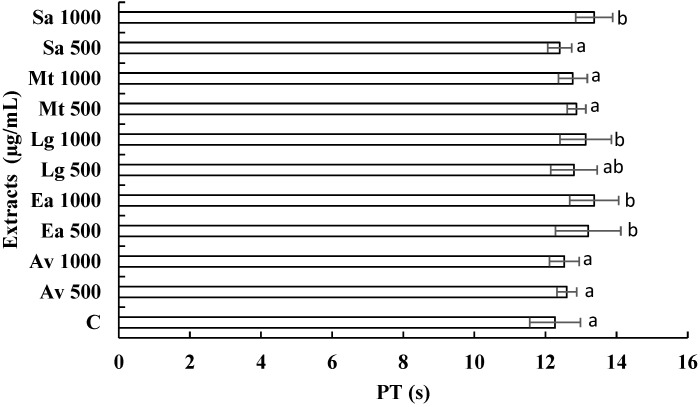
In vitro evaluation of the prothrombin time (PT) caused by *A. vera* (Av), *E. arvense* (Ea), *L. graveolens* (Lg), *M. tenuiflora* (Mt), and *S. aromaticum* (Sa) extracts at 500 and 1000 µg/mL. Data represent means in seconds (s) taken to control a healthy patient’s times (C). Differences between treatments were determined using Tukey’s post hoc test. Significant differences between treatment groups are represented by different letters, and equal letters indicate no significant difference between treatments.

**Figure 2 ijerph-19-07927-f002:**
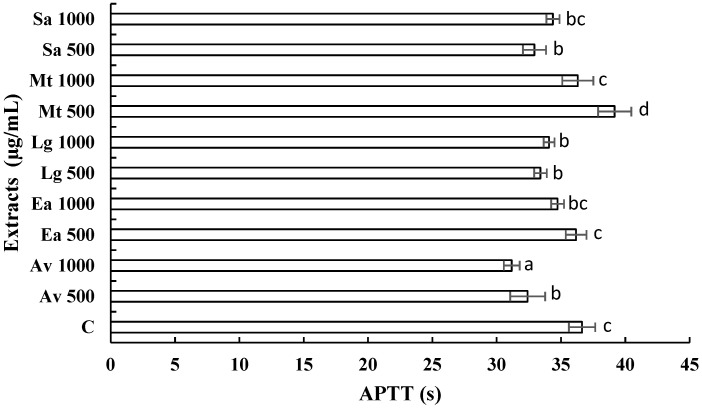
In vitro evaluation of the activated partial thromboplastin time (APTT) caused by *A. vera* (Av), *E. arvense* (Ea), *L. graveolens* (Lg), *M. tenuiflora* (Mt), and *S. aromaticum* (Sa) extracts at 500 and 1000 µg/mL. Data represent means in seconds (s) taken to control a healthy patient’s times (C). Differences between treatments were determined using Tukey’s post hoc test. Significant differences between treatment groups are represented by different letters, and equal letters indicate no significant difference between treatments.

**Table 1 ijerph-19-07927-t001:** Plants used.

Family	Plant Species	Common Name	Part Used
Xanthorrhoeaceae	*Aloe vera* (L.) Burm.f.	Aloe or Sabila	Leaves
Equisetaceae	*Equisetum arvense* L.	Horsetail	Aerial part
Leguminosae	*Mimosa tenuiflora* (Willd.) Poir.	Tepezcohuite	Bark
Verbenaceae	*Lippia graveolens* Kunth.	Mexican oregano	Leaves
Myrtaceae	*Syzygium aromaticum* (L.) Merr. and L.M.Perry.	Clove	Buttons

**Table 2 ijerph-19-07927-t002:** Phytochemical tests.

Test	Chemical Groups	*A. vera*	*E. arvense*	*M. tenuiflora*	*L. graveolens*	*S. aromaticum*
KMnO_4_	Unsaturations	+	+	+	+	+
Antrone	Carbohydrates	+	+	+	+	−
Lieberman–Burchard	Sterols, triterpenes	−	+	+	+	+
Shinoda	Flavonoids	+	−	+	+	+
NaOH	Coumarins	−	+	−	+	+
Baljet	Sesquiterpene lactones	−	−	+	+	+
Sulfuric acid	Quinones	−	−	+	+	+
NaHCO_3_	Carboxyl group	+	−	−	−	−
Ferric chloride	Tannins	−	−	+	+	+
Dragendorff	Alkaloids	−	−	+	−	−
Yield %		11.16	5.23	17.84	8.81	7.20

+ Positive reaction.; − Negative reaction.

**Table 3 ijerph-19-07927-t003:** Activity of ethanol extracts against *S. mutans*.

	Plant Extracts vs. *S. mutans* (Zone of Inhibition in mm)				
Concentration (µg/mL)	*A. vera*	*E. arvense*	*M. tenuiflora*	*L. graveolens*	*S. aromaticum*
250	10.67 ± 0.88	15.33 ± 1.53	15.52 ± 0.33	25.33 ± 1.05 **	16.33 ± 0.58
500	13.67 ± 0.58	16.67 ± 3.06	16.00 ± 2.65	26.00 ± 0.41 **	17.00 ± 1.00
1000	16.00 ± 1.00	20.00 ± 2.00 *	17.67 ± 0.80 *	26.33 ± 1.15 **	18.00 ± 1.34 *
2000	16.33 ± 0.85	22.67 ± 1.58 **	18.00 ± 1.00 *	27.67 ± 1.85 **	19.67 ± 0.58 *
3000	18.67 ± 1.15 *	26.00 ± 2.04 **	18.33 ± 2.08 *	28.48 ± 1.33 **	28.67 ± 0.98 **
C+	15.10 ± 0.77	15.79 ± 0.58	15.00 ± 1.00	15.00 ± 1.67	16.00 ± 0.81
*p-ANOVA*	>0.05	>0.01	>0.05	>0.01	>0.01

Data are expressed as the mean ± SD of three independent replicates. Significant difference compared with C+: * *p* < 0.05, ** *p* < 0.01.

**Table 4 ijerph-19-07927-t004:** Activity of ethanol extracts against *S. sorbinus*.

	Plant Extracts vs. *S. sorbinus* (Zone of Inhibition in mm)				
Concentration (µg/mL)	*A. vera*	*E. arvense*	*M. tenuiflora*	*L. graveolens*	*S. aromaticum*
250	11.00 ± 1.05	8.77 ± 0.41	10.52 ± 1.80	15.67 ± 0.08	10.52 ± 0.05
500	16.67 ± 1.08	12.00 ± 0.89	14.00 ± 0.37	18.00 ± 1.03 *	15.67 ± 0.50
1000	18.00 ± 1.11 *	13.67 ± 1.58	15.67 ± 1.18	19.33 ± 1.53 *	16.00 ± 0.46
2000	18.67 ± 1.53 *	15.76 ± 0.85	16.33 ± 0.58	23.67 ± 2.33 **	19.41 ± 0.58 *
3000	19.67 ± 0.58 *	16.00 ± 1.14	19.67 ± 0.13 *	25.67 ± 0.82 **	23.67 ± 1.53 **
C+	15.10 ± 0.77	15.79 ± 0.58	15.00 ± 1.00	15.00 ± 1.67	16.00 ± 0.81
*p-ANOVA*	>0.05	>0.05	>0.05	>0.01	>0.01

Data are expressed as the mean ± SD of three independent replicates. Significant difference compared with C+: * *p* < 0.05, ** *p* < 0.01.

**Table 5 ijerph-19-07927-t005:** Antioxidant activity.

Plant extract	DPPH Scavenging (IC_50_ in μg/mL)
*A. vera*	8.66 ± 1.42 ^b^
*E. arvense*	2.73 ± 0.87 ^a^
*M. tenuiflora*	464.06 ± 8.58 ^e^
*L. graveolens*	19.80 ± 1.37 ^c^
*S. aromaticum*	37.20 ± 2.88 ^d^
C+ (Vitamin C)	2.23 ± 0.42 ^a^

Data are expressed as the mean ± SD (*p* < 0.05). Differences between treatments were determined using Tukey’s post hoc test. Significant differences between treatment groups are represented by different letters, and equal letters indicate no significant difference between treatments.

**Table 6 ijerph-19-07927-t006:** Cellular viability in percentage (%) and IC_50_ for each evaluated extract.

	Plant Extracts				
Concentration (µg/mL)	*A. vera*	*E. arvense*	*M. tenuiflora*	*L. graveolens*	*S. aromaticum*
125	99.00 ± 0.11 ^a^	89.42 ± 3.52 ^d^	91.11 ± 3.71 ^b^	84.60 ± 3.21 ^b^	88.04 ± 1.36 ^d^
250	99.00 ± 2.28 ^a^	81.68 ± 4.34 ^c^	86..99 ± 5.69 ^b^	79.53 ± 3.50 ^b^	75.89 ± 0.42 ^c^
500	99.04 ± 3.29 ^a^	61.12 ± 1.40 ^b^	58.07 ± 3.72 ^a^	72.31 ± 0.81 ^ab^	50.71 ± 0.34 ^b^
1000	96.71 ± 3.90 ^a^	8.03 ± 1.08 ^a^	55.95 ± 4.52 ^a^	62.50 ± 2.26 ^a^	6.10 ± 1.02 ^a^
C−	100.00 ± 0.00 ^a^	100.00 ± 0.00 ^e^	99.32 ± 0.65 ^c^	100.00 ± 0.00 ^c^	97.26 ± 0.92 ^e^
IC_50_ (μg/mL)	>3000	560.62	1037.43	1384.70	497.51
LL	ND	510.51	953.25	1217.02	449.46
UL	ND	590.42	1161.09	1552.65	550.93

Data are expressed as the mean ± SD (*p* < 0.05). Differences between treatments were determined using Tukey’s post hoc test. Significant differences between treatment groups are represented by different letters, and equal letters indicate no significant difference between treatments. ND: Not determined.

**Table 7 ijerph-19-07927-t007:** Anti-inflammatory activity.

		IL Levels (pg/mL)		
Extract	Hr	TNF-*α*	IL-1 *β*	IL-10
*A. vera*	0	128.35	303.17	85.40
	24	0.00	154.83	19.40
	48	0.00	219.83	0.00
	72	0.00	251.50	0.00
*p-ANOVA*		>0.001	>0.05	>0.001
*E. arvense*	0	39.10	105.67	62.90
	24	0.00	156.36	20.90
	48	68.60	181.08	85.40
	72	222.62	410.25	138.40
*p-ANOVA*		>0.001	>0.01	>0.01
*M. tenuiflora*	0	378.63	21.92	0.00
	24	376.10	64.42	0.00
	48	156.60	100.25	24.90
	72	7.66	189.00	60.40
*p-ANOVA*		>0.05	>0.05	>0.01
*L. graveolens*	0	0.00	251.50	17.40
	24	0.00	236.92	28.90
	48	19.11	117.33	76.40
	72	116.10	112.33	97.40
*p-ANOVA*		>0.01	>0.01	>0.05
*S. aromaticum*	0	354.61	179.83	5.90
	24	327.10	156.08	0.00
	48	304.35	132.75	26.40
	72	239.60	110.25	0.00
*p-ANOVA*		>0.05	>0.05	>0.01

Data are expressed as the mean (*p* < 0.05) of the Interleukin (IL) expression levels in pg/mL.

## Data Availability

The datasets generated and/or analyzed during the present study are available from the corresponding author on reasonable request.

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
