# Peer review of "Preliminary Study of the Antimicrobial, Anticoagulant, Antioxidant, Cytotoxic, and Anti-Inflammatory Activity of Five Selected Plants with Therapeutic Application in Dentistry"

_ijerph, 2022, doi:10.3390/ijerph19137927_

Round 1

Reviewer 1 Report

  1. Please improve the English and check the spelling. Somewhere meaning of the sentence is not clear.
  2.  Add the suggested references and can add more recent references.
  3. improve the discussion part of the manuscript.
  4. The conclusion of the manuscript should be more centered on the selected plants and future prospects.

Reviewer 2 Report

The manuscript written by Lopez Villarreal et all is an extensive material devoted to the introduction of plant extracts in dentistry. The work is carefully prepared in terms of editing. 

It deserves to be published in IJERPH, however, there are several points in which improvements should be made to strengthen the manuscript. 

In the introduction it would be beneficial to add some information about the pharmacological properties of the five selected medicinal plants and to mention why were these species selected to the detriment to others. 

Line 94 - How was the plant material prepared for extraction? Did you use any mills, sieves or other instruments?

Line 97 - The stored extract was dry or liquid?

Line 103 - I suggest to add references that directly describe the mentioned tests;

Line 153 - Please add the international standard ISO to the references;

Lines 220-222 - In order to increase the clarity of the statement regarding S.aromaticum extract, please add the values in brackets as you have done between lines 225 - 229;

Line 264 - Please check whether IC50 for vitamin C is 1,23 μg/mL or 2,23 μg/mL (in correlation to Table 5);

Line 270 - "flower" is written twice in the enumeration of plant organs;

Lines 300 - 303 Regarding IL-1β, A.vera extract triggers a nuanced response, different than E.arvense and M.tenuiflora, I suggest to point out this behaviour;

Lines 314-322   It would be interesting to mention whether any of the five plants has distanced itself from the others in terms of positive or negative effects. Will you continue to study these plants or others?

Reviewer 3 Report

I read the article about Therapeutic Odontological Potential of Five Selected Medicinal 2 Plants Used in Mexico: In vitro Antimicrobial, Anticoagulant, 3 Antioxidant and Anti-inflammatory Properties.

The title is too lengthy and it must be comprehensive. The authors carried out all pharmacological activities in-vitro. How they can co-relate thier results with in-vivo affects??

In Mexico many medicinal plants are using in folk medicines so why the authors selected only five???

The authors said "The purpose of the study was to evaluate the phytochemical composition". The phytochemistry of all selected plants is already known and present in literature so why the authors analyzed it??

DPPH ant-oxidant assay is very basic, why they did not carried the most authentic assays i.e. ferric oxide assay.?

The absorbance (Abs) was noted at 517 nm however the plant extracts contain numerous constituents so this lambda max was for which constituent?

The most abundant anti-inflammatory IL is IL-6 while the authors did not check this marker. 

Why the authors used Tukey post-hoc test?

Reviewer 4 Report

Abelardo Chaves Montes and co-workers, in an article titled Therapeutic Odontological Potential of Five Selected Medicinal Plants Used in Mexico: In vitro Antimicrobial, Anticoagulant, Antioxidant and Anti-inflammatory Properties. » described the thearapeutic odontological potential of selected plants for various properties in vitro.

It is a very interesting study because the use of natural molecules instead of drugs seems, of course, more promising. However, the authors should explain, in the introduction, what led them to choose these 5 plants among those, which are reputed to have the activities required for the field treated.

In addition, several errors and inaccuracies require explanations and corrections. Some of them are listed below but the entire article should be proofread carefully to eliminate small errors before publication.

Line 96 : Finally, the extracts were filtered, rotaevaporated, and the residue stored
(4°C) in amber bottles until use. This sentence will be better as follow : Finally
, the extracts were filtered, MeOH was removed under reduced pressure and stored
(4°C) in amber bottles until use.

Line 100 : KMnO4 must be written KMnO4. It is the samething in table 2

Line 102 : NaHCO3 must be written NaHCO3

Line 239 : « were evaluated, The results….. » must be change to : « were evaluated. The results…. »

Line 287 : in Table 6. ……and IC50 ́s determined for each evaluated extract. » must be change to : « ….. and IC50  is determined for each evaluated extract. »

In Figures 1 and 2 letters a, b, ab, bc, d, must be explicited it is not clear.

Line 367 ref 10 put all the names and not « et al. » ; idem line 374 ref 13

Line 460 : ref 45 : Yamamoto-Furushoa, K.. Expresión de la interleucin : one dot must be removed.

Round 2

Reviewer 1 Report

Revision is satisfactory

Author Response

Dear reviewer, we appreciate your comments and your valuable time spent on our paper.

Reviewer 3 Report

I read the revision and satisfied with it.

Author Response

Dear reviewer, we appreciate your comments and valuable time spent on our paper.